# STXBP6 Gene Mutation: A New Form of SNAREopathy Leads to Developmental Epileptic Encephalopathy

**DOI:** 10.3390/ijms242216436

**Published:** 2023-11-17

**Authors:** Mirella Vinci, Carola Costanza, Rosanna Galati Rando, Simone Treccarichi, Salvatore Saccone, Marco Carotenuto, Michele Roccella, Francesco Calì, Maurizio Elia, Luigi Vetri

**Affiliations:** 1Oasi Research Institute-IRCCS, 94018 Troina, Italy; mvinci@oasi.en.it (M.V.); rgalati@oasi.en.it (R.G.R.); streccarichi@oasi.en.it (S.T.); melia@oasi.en.it (M.E.); lvetri@oasi.en.it (L.V.); 2Department of Psychology, Educational Science and Human Movement, University of Palermo, 90141 Palermo, Italy; carola.costanza@unipa.it (C.C.); michele.roccella@unipa.it (M.R.); 3Department Biological, Geological and Environmental Sciences, University of Catania, Via Androne 81, 95124 Catania, Italy; saccosal@unict.it; 4Clinic of Child and Adolescent Neuropsychiatry, Department of Mental Health, Physical and Preventive Medicine, University of Campania “Luigi Vanvitelli”, 80131 Naples, Italy; marco.carotenuto@unicampania.it

**Keywords:** *STXBP6* gene, epilepsy, next-generation sequencing, SNAP25, amysin

## Abstract

Syntaxin-binding protein 6 (STXBP6), also known as amysin, is an essential component of the SNAP receptor (SNARE) complex and plays a crucial role in neuronal vesicle trafficking. Mutations in genes encoding SNARE proteins are often associated with a broad spectrum of neurological conditions defined as “SNAREopathies”, including epilepsy, intellectual disability, and neurodevelopmental disorders such as autism spectrum disorders. The present whole exome sequencing (WES) study describes, for the first time, the occurrence of developmental epileptic encephalopathy and autism spectrum disorders as a result of a de novo deletion within the *STXBP6* gene. The truncated protein in the *STXBP6* gene leading to a premature stop codon could negatively modulate the synaptic vesicles’ exocytosis. Our research aimed to elucidate a plausible, robust correlation between *STXBP6* gene deletion and the manifestation of developmental epileptic encephalopathy.

## 1. Introduction

In 2022, the International League Against Epilepsy (ILAE) Task Force on Nosology and Definitions divided epilepsy syndromes with onset in neonates and infants into two groups: self-limited epilepsy syndromes that present a spontaneous age-related remission and developmental and epileptic encephalopathies (DEEs), syndromes in which there is a constant impairment of neurodevelopment [1].

DEEs are typically severe neurodevelopmental disorders characterized by recurrent epileptic seizures with neonatal or childhood onset accompanied by psychomotor impairment and subsequently intellectual disability. Typically, seizures are initially drug-resistant and cognitive impairment follows the onset of seizures.

The typical evolution of DEEs has suggested that seizures and interictal EEG abnormalities may contribute to altering the individual’s normal cognitive development. From this perspective, epileptic encephalopathies were defined by the ILAE as the condition in which “epileptic activity itself contributes to severe cognitive and behavioral impairments above and beyond what might be expected from the underlying pathology alone” [2]. However, in light of the growing evidence on the genetic etiopathogenesis of DEEs in 2017, the ILAE suggested using the term “developmental and epileptic encephalopathy” (DEE) to underline that both aspects, genetics and electrical activity, can play a crucial role in clinical presentation [3].

The DEEs include a wide spectrum of manifestations and can be caused by specific gene variants, mutations, or biological factors [4]. Symptoms encompass a spectrum of neuropsychomotor developmental impairments and neurodevelopmental disorders, such as autism spectrum disorders (ASDs), behavioral disorders, and intellectual disability (ID) [3]. The study carried out by Poke et al. [3] on DEE epidemiology emphasized how challenging it is quantifying DEE patients, as many children cannot be classified under any known epilepsy syndromes. However, this study proposed a cumulative DEE incidence rate of 169/100,000 children [5]. Significant challenges influence the care dynamics of caregivers and the overall families of the individuals affected. The severe neurodevelopmental disorders associated with DEEs unquestionably focus the family’s attention and caregiving efforts around the affected family member, resulting in a significant impact in the whole family structure [6].

The genetic landscapes of DEEs are very complex, and an increasing number of new DEE-associated genes have been identified. DEEs can now often be related to pathogenic variants in genes regulating electrical or neurotransmitter-mediated intersynaptic transmission. Moreover, several mutations have the potential to trigger epileptic encephalopathy in certain individuals while causing self-limited epilepsy in others [7,8].

The intricate neural cell crosstalk encompasses numerous signaling pathways regulated via several gene transcriptional processes that modulate the synaptic vesicle cycle. Moreover, astrocytes within glial cells actively modulate the ionic environment, influencing cellular crosstalk in several neural injury conditions, including neurodegenerative disorders [9,10].

SNAREs (soluble N-ethylmaleimide-sensitive factor attachment protein receptor) are a complex family of proteins implicated in synaptic vesicle exocytosis and synaptic transmission. The SNARE complex enables the fusion of the synaptic vesicles with the presynaptic terminals. The role of the SNARE complex, involved in neural communication and activity, is crucial for neurotransmitter release [11,12]. Mutations in genes encoding SNARE proteins have been associated with a broad spectrum of neurological conditions defined as SNAREopathies. These conditions can etiologically belong to multiple syndromes, from simple febrile seizures and infantile spasms to severe early-onset epileptic encephalopathies [13]. Moreover, SNAREopathies also encompass ID, ASD, and movement disorders, although the exact mechanism by which mutations in the SNARE protein cause neurodevelopmental impairment is still largely unknown [14].

ASDs are characterized by early-life developmental anomalies, including deficiencies in synaptogenesis and neurotransmission [15]. Recent advances in sequencing technology have confirmed that the etiology of ASDs is multigenic and highly heterogeneous. Among the candidate genes, the ones related to the SNARE complex (*SNAP25*, *STX1A*, and *VAMP2*) have also been described as pathogenetic for ASDs [16].

The core SNARE complex includes a wide range of proteins, forming a four-helix bundle. This structure consists of two SNAP25 helices (encoded by SNAP25), one syntaxin 1A helix (encoded by the *STX1A* gene), and one synaptobrevin 2 helix (encoded by *VAMP2*) [17]. Causative variants within these genes responsible for SNARE protein synthesis encompass a wide array of genetic alterations, including homozygous splice variants, de novo missense variants, and in-frame deletions affecting only a few amino acids [18].

Genetic variants in genes encoding SNARE complex components, such as SNAP25, have been strongly linked to early-onset developmental epileptic encephalopathy [19]. Mutations in VAMP1/synaptobrevin have also been linked to neurological disorders, as previously reported [20]. As was outlined in previous studies, STX1A is pivotal in calcium-dependent exocytosis and the endocytosis of hormones and neurotransmitters [21]. Furthermore, it is involved in regulating calcium-dependent acrosomal exocytosis in sperm. Additionally, it plays a crucial role in the exocytosis of hormones like insulin or glucagon-like peptide 1 (GLP-1), as indicated by functional similarity [22].

Within the proteins encompassed in the SNARE complex, the syntaxin-binding protein featuring several isoforms is crucial in contributing to the specificity and diversity observed in vesicle trafficking and membrane fusion processes [23,24]. Syntaxin-binding protein six (*STXBP6*) (MIM603215) is essential in regulating SNARE complex formation and neuronal vesicle trafficking. Its protein binds syntaxin-1, syntaxin-4, and SNAP25, forming non-fusogenic complexes that facilitate VAMP2-mediated membrane fusion [23,25,26]. STXBP6 is a 210-amino acid polypeptide containing a unique N-terminus and a C-terminal coiled-coil domain that lacks a hydrophobic stretch that could serve as a transmembrane anchor. It is located in the Golgi apparatus in undifferentiated PC12 cells, regulating trans-Golgi network trafficking and the secretory pathway via its coiled-coil domains [27,28].

While the modification of *STXBP6* is associated with various diseases in different human tissues, including potential roles in neurodegenerative diseases like Alzheimer’s disease (AD), Pick’s disease, and frontotemporal dementia (FTD), its association with DEE in humans has not been previously described [29,30]. To date, no variants within the *STXBP6* gene have been associated with any diseases in humans.

In this study, we reported the first literature-documented case of a DEE associated with a de novo *STXBP6* mutation directly affecting the function of this specific protein.

## 2. Results

### 2.1. Case Description

The 7-year-old girl was the second daughter of healthy, non-consanguineous parents. Intrauterine growth restriction (IUGR) and reduced fetal movements were observed during pregnancy. The baby was born at 33 gestational weeks, and it was delivered via emergency caesarean section due to maternal preeclampsia. The birth weight registered was 1680 g (z-score: −0.665), while the length was 39 cm (z-score: −1.922), the head circumference was 28 cm (z-score: −1.773), and the APGAR score was 8–9.

At birth, the child presented respiratory distress, jaundice, anemia, apnea crisis, retinopathy, and marked hypotonia. Due to the persistence of hypotonia and low weight gain, the newborn underwent an ultrasound brain scan, which exhibited evidence of periventricular hyperechogenicity and bilateral slight dysmorphism showing asymmetry of the frontal horns of the lateral ventricles. The following metabolic analyses were within normal limits: plasma and urine amino acids, chitotriosidase, acetylcarnitine, and urinary mucopolysaccharides dosages.

From an early age, it was observed that sucking problems led to poor weight gain and difficulties in feeding. From the age of 12 months, the girl underwent a rehabilitation path of physio kinesiotherapy, psychomotor treatment, and speech therapy. At around 16 months, the clinical situation of the child was characterized by growth delay, with a weight of 6.8 kg (z-score: −2.799), a length of 69.5 cm (z-score: −2.933), and a head circumference of 41.5 cm (z-score: −1.558). Lallation was absent; eye contact was present. Generalized muscle hypotonia and ligamentous hyperlaxity were detected in the overall body assessment. Reflexes were normal and bilaterally symmetrical in both extremities, and clonus was absent. Good head control, trunk control in acquisition, and poor hand–eye coordination skills were observed. During eye examination, an alternating esotropia condition and the excavation of the papilla were detected in both the right and left eyes. The thyroid-stimulating hormone level was 9.420 µU mL^−1^, while the free thyroxine showed normal values, indicating primary hypothyroidism. Therefore, a treatment based on levothyroxine was started.

At 19 months, the child suffered from two episodes of simple febrile seizures. At 22 months, the child recorded a weight of 7.8 kg (z-score: −2.702), a length of 77 cm (z-score: −2.001), and a head circumference of 42.5 cm (z-score: −2.983). All the analyzed anthropometric data are summarized in Table 1. During this period, monthly seizures occurred, characterized by loss of consciousness, staring eyes, and snoring sounds. Consequently, a treatment based on valproic acid was started with progressive dosage increments without achieving seizure-free status (40 mg twice a day; 80 mg twice a day; 125 mg twice a day; and 150 mg twice a day).

Visual potentials evoked via flash and pattern stimulation exams showed regular morphology, reduced amplitude, and increased latency time. The magnetic resonance imaging (MRI) performed at 22 months revealed microcephaly, a simplified cortical spinning pattern, mild ventriculomegaly, and periventricular white substance thinning (Figure 1). A slight delay in myelination was also detected, which was related to regions of altered hyperintense signal in FLAIR in bilateral periventricular with extension to bilaterally radiated crowns. The imaging also detected hypoplasia of the corpus callosum and anterior commissure, in addition to dysmorphism of the diencephalic–mesencephalic junction, with a partial fusion between the midbrain and hypothalamus/anterior optic tracts, as well as a small median cleft.

Starting at 24 months, documented generalized tonic–clonic seizures appeared together with perioral cyanosis, manifesting as a bluish discoloration of the distal extremities. Consequently, a therapy regime with levetiracetam was started as an additional treatment, determining several years of freedom from seizures.

At around the age of two, the electroencephalogram (EEG) recorded showed slow background activity without evident epileptiform abnormalities within the normal limits concerning morphology, amplitude, and latency time.

At approximately 30 months, the weight was 8.5 kg (z-score: −3.401) and the length was 83 cm (z-score: −2.175), while the body mass index (BMI) was 12.3 kg/m (z-score: −2.908). The clinical picture at this age showed a potential and inconstant lack of eye contact, ligamentous hyperlaxity, and axial and segmental muscle hypotonia. It was also observed that autonomous transitional movements occurred in postural passages, including the assumption of all four positions. With assistance, the child could stand upright and take a few steps. Dysmorphic features were observed, such as epicanthus, broad nasal root, wide auricles, periorbital fullness, wide mouth, full lips, tapered fingers, and long and thin feet (Figure 2).

From the age of four, sporadic seizures occurred with a frequency of about one episode per year, often triggered by fever and characterized by eye fixity, break of contact, hypertonia, cyanosis, and subsequent hypotonia (with an average of 30 s). The 7-year EEG showed sharp waves and short sequences of fast activity over the temporal regions of the two hemispheres (Figure 3). At the age of seven, the clinical phenotype was characterized by unsupported eye-catching, non-response to name, inappropriate laughter, and lack of speech. According to these features, a diagnosis of autism spectrum disorder was made according to the DSM 5 criteria [31]. Autonomous gait was present, even if uncertain and broadly based (acquired by about two years). Ligamentous hyperlaxity, reduced tone and trophism, fascination for water, motor stereotypes, and echolalia were also present. Considering the girl’s clinical picture, karyotype (46XX), array CGH, and molecular analysis of critical regions for Prader–Willi and Angelman syndrome were performed, and the results were all normal; therefore, whole exome sequencing (WES) was carried out.

### 2.2. NGS

NGS analysis revealed the presence of a novel heterozygous de novo variant, specifically c.313_323delGAAAATGCTTT within the *STXBP6* gene (NM_014178.8) (Figure 4a). Sanger sequencing on the family members (patient, father, and mother) confirmed the presence of this mutation in the child affected by a DEE (Figure 4b). Remarkably, this deletion leads to the generation of a premature termination codon at p.Glu105Ter, resulting in the truncation of the protein (Figure 4c).

The prediction showed a notable variation in the polypeptide length, variating from the standard 210 amino acids to a modified length of 105 amino acids. As a result, the modified protein lacked the domain “N-terminal pleckstrin homology” (amino acids 151–210) (Figure 5a).

The novel mutation detected was not found in the HGMD Professional Database 2023.2 (www.hgmd.cf.ac.uk) (accessed on 10 July 2023), 1000 Genomes Database, Exome Aggregation Consortium, or GnomAD.

## 3. Discussion

To date, no variants within the *STXBP6* gene have been associated with any diseases in humans. In the present study, we have, for the first time, associated the *STXBP6* gene in a patient diagnosed with a DEE. Therefore, as expected, the Mendelian Inheritance in Man (MIM) number has not been assigned yet. The deletion detected (c.313_323delGAAAATGCTTT) in the gene *STXBP6* leads to the generation of the premature stop codon (p.Glu105Ter), thus causing the loss of the “N-terminal pleckstrin homology” domain. Amisyn (*STXBP6* gene) acts as a negative regulator of the SNARE complex, modulating exocytosis [25]. The SNARE complex is a composite network of proteins, and disturbances in any of its components can bring about diverse effects on cellular function and contribute to different pathological conditions. The SNARE complex is involved in vesicle fusion, neurotransmitter release in neurons, and membrane trafficking and fusion in other cells [12,33]. Consequently, a modification similar to the one we detected in the three-dimensional structure folding of a SNARE complex protein can significantly impact this process.

Several neurological and neurodevelopmental disorders have been associated with mutations in the SNARE complex proteins due to the fundamental involvement of the complex in cortical synaptic growth in the first months of life. Variants in the *STX1B* gene contribute to many distinct epilepsy phenotypes [34]. Mutations in the syntaxin-1A (*STX1A*) gene have been linked to early-onset epileptic encephalopathy [35]. Mutations in this gene can disrupt synaptic vesicle release and neurotransmitter signaling, leading to seizures and cognitive impairments.

In addition to STXBP6, numerous STXBPs also interact with syntaxin (STX) in regulating the SNARE complex during the exocytosis process. Despite their distinct three-dimensional structure, they share the main function of binding to STX1 and assembling into the SNARE complex [26,36]. These proteins are, for example, STXBP1A, STXBP1B, and STXBP4. They are also associated with DEEs, as described by numerous studies [37,38,39,40]. Based on the previously mentioned studies, we can hypothesize that functional modifications in the amino acid sequences of different *STXBP* genes may be associated with a similar clinical presentation characterized by epilepsy. For example, the work of Yamashita et al. (2016) [41] revealed a nonsense mutation in the *STXBP1* gene related to a 50% reduction in mRNA and protein expression levels in comparison to the control neurons. This nonsense mutation caused a DEE in the patient. Grone et al.’s (2016) research on zebrafish mutants [42], through CRISPR/Cas9 gene editing, further supports our study. They observed severe effects in homozygous *stxbp1a* mutants, including immobility, reduced brain activity, and early mortality, while homozygous *stxbp1b* mutants displayed spontaneous seizures. Saitsu et al. (2008) [43] also found a de novo 2.0Mb microdeletion in the *STXBP1* gene associated with DEEs using array comparative genomic hybridization (array CGH).

As previously reported [26], STXBP6 directly interacts with proteins (STX4) implicated in synaptic vesicles and the molecular regulation of neurotransmitter release, playing a pivotal role in the synaptic function of specific neuronal systems (SNAP25). SNAP25 was involved in ASD, seizures, and ID [44]. SNAP25 is a component of the trans-SNARE complex, responsible for governing the specificity of membrane fusion by forming a tight complex that brings the synaptic vesicle and plasma membranes together [45] (Figure 5a). Therefore, the STRING analysis revealed a robust interaction between the *STXBP6* and *SNAP25* genes, reinforcing the well-documented association of *SNAP25* with autism. As extensively reported in the literature, polymorphisms in the *SNAP25* gene are intricately related to ASDs [46,47,48].

Moreover, in a study conducted on a patient showing both an ASD and coloboma [49], a mosaicism involving the formation of a ring chromosome was identified. The detected mosaicism included an inverted duplication of the proximal region of chromosome 14, encompassing the *STXBP6* gene. This chromosomal aberration resulted in the silencing of the *STXBP6* gene, ultimately triggering the development of ASD. Similarly, in a multi-omics data integration study [50], *STXBP6* was identified as a candidate gene for ASD.

In our patient, the presence of a severe DEE with a comorbid ASD led us to suppose that this clinical picture is linked with the pathogenic variant identified in the *STXBP6* gene.

However, further investigations are needed, and a larger cohort of patients must be identified to gain a more comprehensive clarification of the phenotype associated with *STXBP6* gene-related disorders. Furthermore, functional assays will provide insights into the precise consequences of *STXBP6* mutation.

## 4. Materials and Methods

### 4.1. Libraries Preparation and NGS Analysis

Genomic DNA was isolated from peripheral blood leukocytes obtained from the clinical case, as well from the father and the mother. The extraction protocol applied was a non-organic and non-enzymatic extraction method developed by Lahiri et al., 1992 [51]. Exome analysis was performed using the Ion AmpliSeq™ Exome RDY kits, following the manufacturer’s instructions (Thermo Fisher Scientific, Waltham, MA, USA). The quality of libraries was assessed using DNA 1000 chips on the Tape Station 4200 (Agilent, Santa Clara, CA, USA) and Qubit dsDNA BR Assay kits (Invitrogen, Waltham, MA, USA). Template preparation, clonal amplification, recovery, and enrichment of template-positive Ion Sphere™ particles and loading of sequencing-ready Ion Torrent semiconductor chips were performed with the Ion Chef™ system (Thermo Fisher Scientific, Waltham, MA, USA). Finally, we sequenced each loaded Ion 550™ chip on the S5 system (Thermo Fisher Scientific, Waltham, MA, USA). Overall, 98% of regions of interest have a minimum coverage of at least 20X. Data of runs were processed using the Ion Torrent Suite 5.16, Variant Caller 5.16, Coverage Analysis 5.16 (Thermo Fisher Scientific, Waltham, MA, USA), Ion Reporter (Thermo Fisher Scientific, Waltham, MA, USA), and/or wANNOVAR tools [52]. DNA sequences were displayed using Integrated Genomics Viewer [53].

We confirmed pathogenic variants via conventional Sanger sequencing (Applied Biosystems Prism 3130 DNA Analyzer, Thermo Fisher Scientific, Waltham, MA, USA).

### 4.2. Data Analysis

Data analysis was performed to assess the structure of the STXBP6 protein using the Uniprot database (https://www.uniprot.org/) (accessed on 10 July 2023). Specifically, the Uniprot model of the STXBP6 protein is shown in Figure 5a. Additionally, Figure 5b reports the secretory vesicle exocytosis model developed by Kondratiuk et al. (2020) [32].

On the other hand, for the data analysis of the sequence, we excluded all the common variants and non-exonic polymorphisms, keeping polymorphisms with a minor allele frequency (MAF) of <1% in the public databases gnomAD Exomes, 1000 Genome Project, and Exome Sequencing Project. The pathogenic variants were searched in the Human Gene Mutation Database (HGMD Professional 2023). Specifically, we used the VarAFT filtering and annotation tool (https://varaft.eu/) on vcf files (accessed on 10 July 2023). The variation that we found was classified according to the “American College of Medical Genetics” (ACMG) guidelines [54], and it was performed with VarSome in accordance with Kopanos et al. (2019) [55] and other literature evidence [56]. The evidence of pathogenicity was assigned to each variant identified as follows: benign, likely benign, uncertain significance, likely pathogenic, and pathogenic.

The model for understanding the interaction between the STXBP6 protein and other SNARE proteins was developed utilizing the STRING database (https://string-db.org/) (accessed on 10 July 2023) to construct a functional protein association network. This approach allowed for a comprehensive exploration of potential interactions, enhancing our understanding of the roles played by STXBP6 in intracellular vesicle dynamics and synaptic membrane processes.

## 5. Conclusions

In conclusion, our study presents a novel and compelling case of developmental epileptic encephalopathy and ASD resulting from a de novo deletion within the *STXBP6* gene, responsible for encoding the pivotal syntaxin-binding protein 6 (STXBP6), also known as amysin. These findings provide valuable insights into the potential involvement of the *STXBP6* gene in the pathogenesis of neurological conditions, expanding on the body of evidence on SNAREopathies.

This report not only aids in advancing our comprehension of certain forms of DEEs but also underscores the significance of persistent research into the genetic factors underlying early-onset epilepsy. Further investigations are essential to validate the novel gene mutation and gain a deeper understanding of the intricate interaction among genes encoding proteins involved in the exocytosis process.

We aimed at bringing our study to the attention of the scientific community in order to stimulate future functional research, which may elucidate the pathogenic mechanisms of the *STXBP6* gene. This could ultimately confirm the role of the *STXBP6* gene in the etiology of developmental and epileptic encephalopathy (DEE). Moreover, the identification of more patients with *STXBP6* gene mutations could facilitate a more precise delineation of the genotype–phenotype correlation.

## Figures and Tables

**Figure 1 ijms-24-16436-f001:**
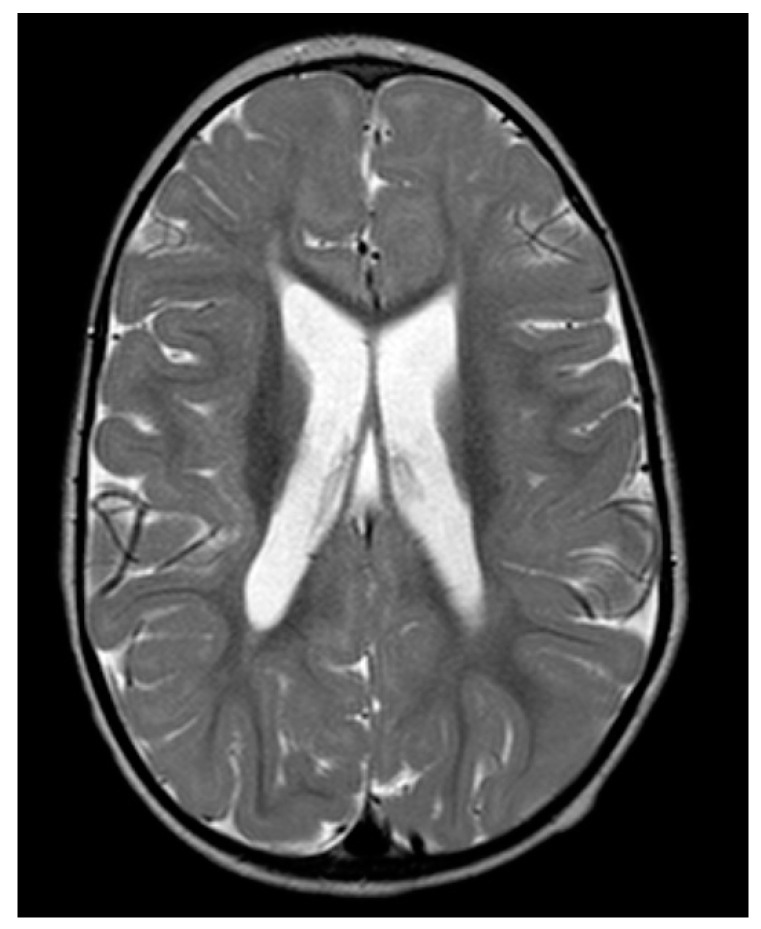
An axial MRI T2 image at 22 months shows a simplified cortical spinning pattern, mild ventriculomegaly, and periventricular white substance thinning.

**Figure 2 ijms-24-16436-f002:**
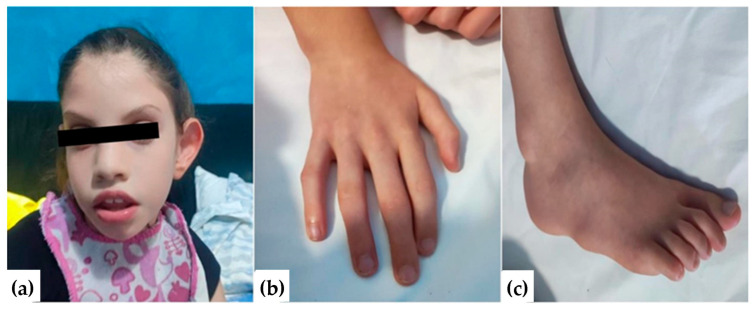
Dysmorphic features of the patient with a *STXBP6* mutation. (**a**) Broad nasal root, wide auricles, periorbital fullness, wide mouth, and full lips. (**b**) Tapered fingers. (**c**) Long and thin feet.

**Figure 3 ijms-24-16436-f003:**
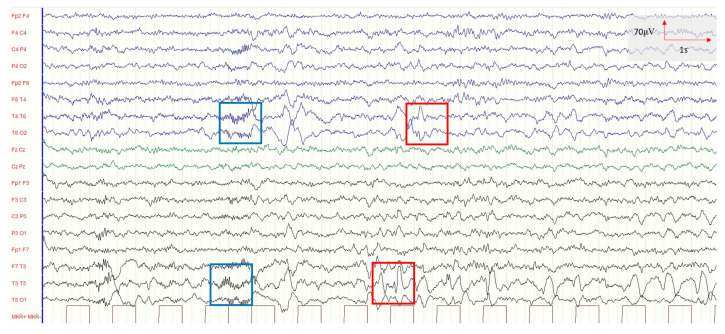
Electroencephalogram (EEG) showing sharp waves (red rectangles) and short sequences of fast activity (blue rectangles) over the temporal regions (T4–T6 and T3–T5) of the two hemispheres.

**Figure 4 ijms-24-16436-f004:**
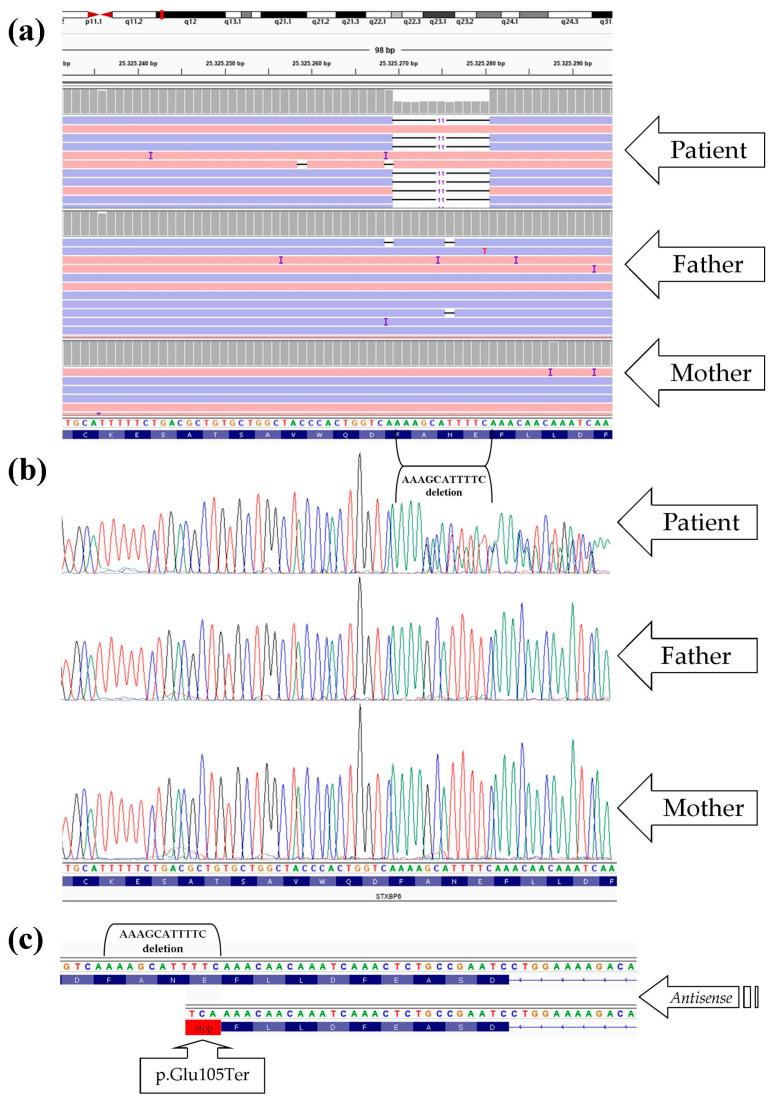
(**a**) Next-generation sequencing (NGS) visualization with the IGV (Integrative Genomics Viewer) of the deletions of GAAAATGCTTT nucleotides (reverse) in the *STXBP6* gene. Pink and blue reads are aligned to the forward strand and reverse strand, respectively. Reads total count: patient = 123 (AAAGCATTTTC = 48 reads; deletion = 54 reads); father = 77; and mother = 123. (**b**) Sanger sequencing displaying the frameshift of the reading frame observed in the patient. Parents exhibited the wild-type alleles, confirming the de novo mutation detected in the clinical case. (**c**) In the patient, de novo heterozygous deletion results in a frameshift and premature termination codon (p.Glu105Ter).

**Figure 5 ijms-24-16436-f005:**
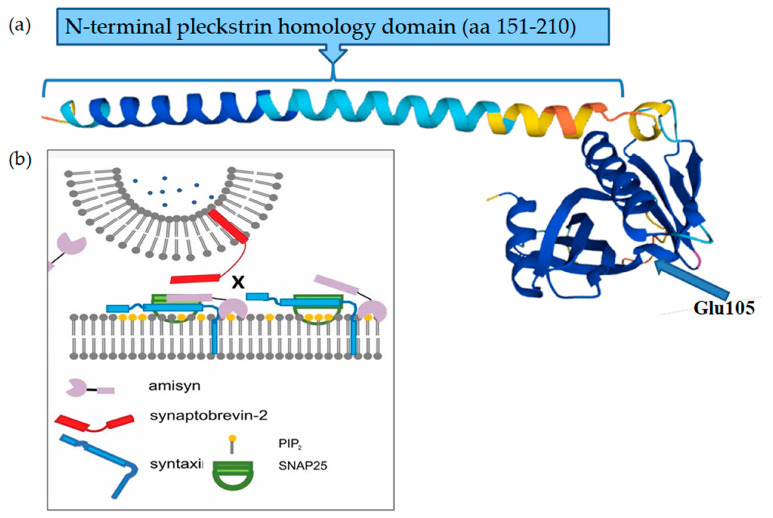
(**a**) In silico prediction model of the protein structure of syntaxin-binding protein 6 (amisyn) generated via the Uniprot database. Deletions c.313_323delGAAAATGCTTT in the *STXBP6* gene (NM_014178.8) generate a premature termination codon at p.Glu105Ter. The truncated protein exhibited a significant reduction in the amino acids in the polypeptide chain (105 versus 210), with the absence of the “N-terminal pleckstrin homology” domain (aa 151–210). The missing branch represented by the truncated domain significantly affects the fusion with the vesicle and thereby with the plasma membrane. (**b**) Model of the role of amisyn in secretory vesicle exocytosis (modified by Kondratiuk et al., 2020) [32]. Amisyn acts as a negative regulator of the SNARE complex by competing with the fusion-active synaptobrevin-2. Amisyn contains a N-terminal pleckstrin homology domain (deleted in the patient) that mediates its transient association with the plasma membrane, modulating the entire exocytosis process.

**Table 1 ijms-24-16436-t001:** Anthropometric measurements at different ages.

Life Stage	Weight	Length	Head Circumference
At birth	1680 g	39 cm	28 cm
At 16 months	6.8 kg	69.5 cm	41.5 cm
At 22 months	7.8 kg	77 cm	42.5 cm

## Data Availability

Data are contained within the article.

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
