# Peer review of "STXBP6 Gene Mutation: A New Form of SNAREopathy Leads to Developmental Epileptic Encephalopathy"

_ijms, 2023, doi:10.3390/ijms242216436_

Round 1

Reviewer 1 Report

Comments and Suggestions for Authors

Abstract:

Line 17 - typo "ant" should be and

Line 21 - is "comorbi" a word or typo?

Line 22 - (de novo deletion, within the STXBP6 gene.The production) add a space between the two words

Line 24 - (exocytosisof syna...) fix the typo

Line 94 -  

  1. Full-length STXBP6 bound syntaxin-1 more efficiently than the coiled-coil domain alone. 

  2. I don't know what is being said here. 

Introduction:

Does not provide clear background on the study and what the gap in the literature is and how the authors aim to fill in the gap with their research. The introduction needs to be redone with proper grammar and spelling. 

Methods: 

Not enough details are described for the family, genetic history of diseases, type of genomic DNA extraction kit used etc. Line 300 - 301 I have no idea what is being said. I think that more data on the whole genome sequencing aspect could be included to support the findings in this study. More details are required for this section. It is not easy to follow what the authors did. Authors may want to re-write their results and divide it into methods and results because most of the methods are in the results section, please review this entire setion and the results secion and re-write.

Results: 

Why is the Sanger sequencing data not show? 

Discussion: 

Well written and I am able to follow what is being explained. Ties well with the Introduction and Results. There a few spelling and grammatical errors where spaces need to be added between the citation reference number and the word that follows - check attachment with highlights in yellow.

Conclusion:

I definitely agree with the below but what is the take home message from this study? 

  1. Further investigations are essential to validate the novel

  2. 327  gene mutation and gain a deeper understanding of the intricate interaction among genes

  3. 328  encoding proteins involved in the exocytosis process. 

Comments on the Quality of English Language

There are major grammatic and spelling errors in this manuscript which needs to be redone and rechecked by an expert in English. I could not comprehend anything in this manuscript at all becuase of the poor English.

Author Response

REVIEWER 1

Abstract:

Line 17 - typo "ant" should be and

R:  Thanks for the suggestion. I corrected as you indicated.

Line 21 - is "comorbi" a word or typo?

R:  Thanks for the suggestion. It was a typo but we preferred to change in the abstract the word “comorbid” with “and”.

Line 22 - (de novo deletion, within the STXBP6 gene.The production) add a space between the two words

R: Thanks for the suggestion. The space was inserted as required.

Line 24 - (exocytosisof syna...) fix the typo

R: Thanks for the suggestion. The space was inserted as required

Line 94 -  

Full-length STXBP6 bounds syntaxin-1 more efficiently than the coiled-coil domain alone.
I don't know what is being said here. 

R: Thanks for the suggestion. The sentence was removed and the role of STXBP6 was well explained in the previous paragraph.

Introduction:

Does not provide clear background on the study and what the gap in the literature is and how the authors aim to fill in the gap with their research. The introduction needs to be redone with proper grammar and spelling. 

R: The introduction was deeply revised. Particularly, we remarked, as suggested, the current gap in the literature. I hope that our case report can attract the attention of scientific community and produce more researches about the role of STXBP6 within SNAREopathy.

Methods: 

Not enough details are described for the family, genetic history of diseases, type of genomic DNA extraction kit used etc. Line 300 - 301 I have no idea what is being said. I think that more data on the whole genome sequencing aspect could be included to support the findings in this study. More details are required for this section. It is not easy to follow what the authors did. Authors may want to re-write their results and divide it into methods and results because most of the methods are in the results section, please review this entire setion and the results secion and re-write.

R: We inserted in “materials and methods” section the reference related to the DNA extraction protocol which was developed by Lahiri et al. 1992. Furthermore, the sentence at 300-301 line was deeply revised specifying the exocytosis model used and reported in Figure 4b. On the other hand, we inserted additional details regarding the NGS analysis, as requested. The methodology inserted in the Result section was moved into the materials and methods section, as suggested.

Results: 

Why is the Sanger sequencing data not show? 

R: Thanks for the suggestion. Electropherogram of Sanger sequencing was added.

Discussion: 

Well written and I am able to follow what is being explained. Ties well with the Introduction and Results. There a few spelling and grammatical errors where spaces need to be added between the citation reference number and the word that follows - check attachment with highlights in yellow.

R: Thanks for the appreciation expressed about the Discussion section, that was also revised as suggested.

Conclusion:

I definitely agree with the below but what is the take home message from this study? 

Further investigations are essential to validate the novel gene mutation and gain a deeper understanding of the intricate interaction among genes encoding proteins involved in the exocytosis process. 

R: In our case report we describe the first case of DEE and autism spectrum disorder related to a mutation of STXBP6 gene, we hope that our first report can stimulate more research study about the role of STXBP6 in the SNAREopathy. We have integrated and explained this concept better in the conclusions.

Reviewer 2 Report

Comments and Suggestions for Authors

In the manuscript titled WES revealed a new SNAREopathy linked to STXBP6 gene causing Developmental Epileptic Encephalopathy and Autism Spectrum Disorders, the authors provided a clinical report of the patient and performed Next Generation Sequencing (NGS) result. This paper attempted to make a robust correlation between STXBP6 gene mutation and the manifestation of developmental epileptic encephalopathy through documenting the patient’s clinical history and detecting the deletion of nucleotides in the STXBP6 gene. This is a pretty good paper for a case study, but for a research article, it may need more effort to confirm a direct relation between STXBP6 and the disease, as protein structure change should be directly observed from patient’s clinical samples, not from a prediction model. Furthermore, the number of samples, which is one in this study, is not enough to support their claim.

Major comments:

1. This manuscript should belong to the case report section, not the research article section as stated above. If the authors insist this should be a research article, please make more efforts to link the DEE with the STXBP6 gene. You may start from providing the real protein structure of the muted gene, and prove the dysfunction of exocytosis of synaptic vesicles with patient data.

2. There are some issues with the title of the manuscript.

Please avoid using abbreviations in the title. I searched the whole article for the WES meaning, only to find its full name in line 180, which makes people confused upon reading this title.

Based on the introduction part, autism spectrum disorder is one of the symptoms under the developmental and epileptic encephalopathy (DEE), so “and” relation is not suitable. Same problem in the abstract part, please address.

There’s also language problem in the title, making it verbose. You can consider using this title: “STXBP6 gene- a new link to SNAREopathy in developmental epileptic encephalopathy.”

3. For the introduction part:

The first sentence is too general and didn’t give much information. “Due to etiology”, which etiology? “superimposed epileptic activity” is a manifestation not a reason.

There are several logic jumps in the introduction part. Line 41-46 mentioned the single gene. Line 47-56 talked about several gene-SNARE complex. Line 69-77 mentioned single gene again.

4. For the result part:

Line 112-113, “treated with erythropoietin” appear among all the symptoms, is the “apnea crisis, retinopathy, and marked hypotonia” the results after treatment? This is confused, please clarify.

Figure 2, please mark the sharp wave and fast activity with rectangles and give the name of its located channel.

There are no 2.2 sections in the result part, it jumped to 2.3 NGS.

Minor comments:

There are many typos in this manuscript, such as line 17 “ant” and line 21 “comorbi”, please proofread the article to find them out and correct them.

Line 179, please give the full name of the abbreviations-CGH.

Comments on the Quality of English Language

The English quality is good but still some typos and expressions need to be addressed and improved.

Author Response

REVIEWER 2

In the manuscript titled WES revealed a new SNAREopathy linked to STXBP6 gene causing Developmental Epileptic Encephalopathy and Autism Spectrum Disorders, the authors provided a clinical report of the patient and performed Next Generation Sequencing (NGS) result. This paper attempted to make a robust correlation between STXBP6 gene mutation and the manifestation of developmental epileptic encephalopathy through documenting the patient’s clinical history and detecting the deletion of nucleotides in the STXBP6 gene. This is a pretty good paper for a case study, but for a research article, it may need more effort to confirm a direct relation between STXBP6 and the disease, as protein structure change should be directly observed from patient’s clinical samples, not from a prediction model. Furthermore, the number of samples, which is one in this study, is not enough to support their claim.

R: Following the suggestion and agreeing with the reviewer, we changed the type of paper to Case Report and we hope that future studies will shed some light on the pathogenic mechanisms of STXBP6 mutation.

Major comments:

  1. This manuscript should belong to the case report section, not the research article section as stated above. If the authors insist this should be a research article, please make more efforts to link the DEE with the STXBP6 gene. You may start from providing the real protein structure of the muted gene, and prove the dysfunction of exocytosis of synaptic vesicles with patient data.

R:  Thanks for the suggestion. According with your comment we modified the type of paper in “Case Report”.

  1. There are some issues with the title of the manuscript.

Please avoid using abbreviations in the title. I searched the whole article for the WES meaning, only to find its full name in line 180, which makes people confused upon reading this title.

R: Thanks for the suggestion. We changed the title

Based on the introduction part, autism spectrum disorder is one of the symptoms under the developmental and epileptic encephalopathy (DEE), so “and” relation is not suitable. Same problem in the abstract part, please address.

R: DEEs are often characterized by neurodevelopmental disorders like autism spectrum disorder but autism is not invariably present. In the abstract we would underline the type of neurodevelopmental disorders characterizing the DEE caused by STXBP6 mutation.

There’s also language problem in the title, making it verbose. You can consider using this title: “STXBP6 gene- a new link to SNAREopathy in developmental epileptic encephalopathy.”

R: According to your suggestion we changed the title in “STXBP6 gene mutation: a new form of SNAREophaty leads to developmental epileptic encephalopathy”

  1. For the introduction part:

The first sentence is too general and didn’t give much information. “Due to etiology”, which etiology? “superimposed epileptic activity” is a manifestation not a reason.

R:  Thanks for the suggestion. We remodulated the sentence.

There are several logic jumps in the introduction part. Line 41-46 mentioned the single gene. Line 47-56 talked about several gene-SNARE complex. Line 69-77 mentioned single gene again.

R: Thanks for the suggestion. We modified the suggested parts in order to clarify the complex genetic landscape of DEE.

  1. For the result part:

Line 112-113, “treated with erythropoietin” appear among all the symptoms, is the “apnea crisis, retinopathy, and marked hypotonia” the results after treatment? This is confused, please clarify.

R: Thanks for the suggestion. Symptoms are not caused by erythropoietin treatment,  the sentence war revised for improving clarity and readability.

Figure 2, please mark the sharp wave and fast activity with rectangles and give the name of its located channel.

R: The sharp wave was marked in Figure 2 as recommended. In addition, the Figure description was reformulated.

There are no 2.2 sections in the result part, it jumped to 2.3 NGS.

R: Thanks for the suggestion. We corrected the error.

Minor comments:

There are many typos in this manuscript, such as line 17 “ant” and line 21 “comorbi”, please proofread the article to find them out and correct them.

R: All the above typos, and others we detected within the text, were corrected.

Line 179, please give the full name of the abbreviations-CGH.

R: Thanks for the suggestion. This was done.

Reviewer 3 Report

Comments and Suggestions for Authors

Vinci et al. "WES revealed a new SNAREopathy linked to STXBP6 gene causing Developmental Epileptic Encephalopathy and Autism  Spectrum Disorder" is an interesting study, where the authors tested the hypothesis about the correlation between STXBP6 gene deletion and the manifestation of developmental epileptic encephalopathy and autism. The authors found the occurrence of developmental epileptic encephalopathy comorbi to autism spectrum disorders as a result of a de novo deletion, within the STXBP6 gene.
Further, the authors also determined that the deletion of this gene resulted in the truncated protein structure that is associated with the exocytosis process.

The strength of the article is that it investigates the mechanism of diseases in human subjects. However, there are weaknesses in the article, which should be properly addressed below.

Abstract section
Please check the grammatical and space issues throughout  the manuscript. For example, Line 17: "ant"; Line 24: "exocytosisof"

Introduction section or discussion sections:
Line 36-40: I encourage the authors to bring CDC or WHO  quantification percentage data affected by DEEs globally.

Line 47: The authors have highlighted the complex neural cell crosstalk. However, it did not provide further details about the role of glia in the cross-talk. For further information, please see PMID 37363320,

Line 48: Check the spacing error "byseveral"

Line 49: Check the grammatical issues 'ranscription The SNARE.

Line 81 and Line 85: The word "Amisyn" is redundant. As you have already introduced Amisyn or STXBP6 is the same thing repeatedly. It is better to remove it.

Method Section: Please rearrange your Method section immediately following the introduction section. In other words, please follow the scientifically accepted right order for writing headings with journal guidelines.
I genuinely think the article lacks a multidisciplinary approach to validate the hypothesis. Molecular biological techniques (Western blot techniques, RT-PCR), biochemical analysis, and other detailed behavior testing might have added more value to the article.  I encourage the authors to think of generating more findings and figures.

Result section:
Line 108: "33o", I do not understand the symbol after 33.

Figures: All figures need more details. For example how the figures are obtained, at what resolution. What are the findings the authors saw in each figure should be mentioned in detail in the figure legends.

Line 117-118: It would be useful to provide those data for each test, although the values are not significant.
Line 156-159: Whenever possible, Presenting the data in the Tables looks much better than this text language.

Figure 2: The scaling introduced herein in Figure 2 overlaps with the original EEG traces. I encourage you not to overlap it with the original traces. You may introduce an arrowhead for those signals that the readers need to focus on to observe the findings. Please also define those square-shaped structures within the figure itself at the bottom of the x-axis if they signify anything.

Conclusion section:
Line 318-319: The opening sentence in the conclusion is not a conclusive statement. Please rewrite more appropriately.

Line 323: "As was reported": Please make sure that this type of grammatical issue is taken care of throughout the manuscript.

I also see at the ending section of the introduction, that there are several one or two-sentence paragraphs, please try to merge those relevant sentences together.

Overall, the manuscript has many grammatical errors and spacing error issues, which must be fixed. Only 4 figures are not sufficient (The first figure might not need to be the patient's face but the list of the symptoms in the box may be useful). The authors need to investigate the research problem using a more interdisciplinary approach and the sample size is very low. This article may be a better fit for the case study.

Author Response

REVIEWER 3

Vinci et al. "WES revealed a new SNAREopathy linked to STXBP6 gene causing Developmental Epileptic Encephalopathy and Autism Spectrum Disorder" is an interesting study, where the authors tested the hypothesis about the correlation between STXBP6 gene deletion and the manifestation of developmental epileptic encephalopathy and autism. The authors found the occurrence of developmental epileptic encephalopathy comorbi to autism spectrum disorders as a result of a de novo deletion, within the STXBP6 gene.

Further, the authors also determined that the deletion of this gene resulted in the truncated protein structure that is associated with the exocytosis process.

The strength of the article is that it investigates the mechanism of diseases in human subjects. However, there are weaknesses in the article, which should be properly addressed below.

Abstract section

Please check the grammatical and space issues throughout  the manuscript. For example, Line 17: "ant"; Line 24: "exocytosisof"

R: All the typos, and others we detected along the text, were corrected.

Introduction section or discussion sections:

Line 36-40: I encourage the authors to bring CDC or WHO  quantification percentage data affected by DEEs globally.

R: Thanks for the suggestion. There are no epidemiologic CDC or WHO data about DEE. We reported data of the recent study of Poke et al (2023) “Epidemiology of Developmental and Epileptic Encephalopathy and of Intellectual Disability and Epilepsy in Children”

Line 47: The authors have highlighted the complex neural cell crosstalk. However, it did not provide further details about the role of glia in the cross-talk. For further information, please see PMID 37363320,

R: Thanks for the suggestion. We added further details on the role of glia and added the relative citation.

Line 48: Check the spacing error "byseveral"

R: Thanks for the suggestion. This was done. This and other similar typos was presumably due to the older version of word software we have used. We apologize for this.

Line 49: Check the grammatical issues 'ranscription The SNARE.

R: Thanks for the suggestion. This was done.

Line 81 and Line 85: The word "Amisyn" is redundant. As you have already introduced Amisyn or STXBP6 is the same thing repeatedly. It is better to remove it.

R: We revised the suggested lines, removing the redundance of the word Amysin.

Method Section: Please rearrange your Method section immediately following the introduction section. In other words, please follow the scientifically accepted right order for writing headings with journal guidelines.

R: We followed the journal guidelines, that put the Materials and Methods section after the Discussion and before the Conclusion. We do not believe we can change the order of sections as intended for the IJMS journal.

I genuinely think the article lacks a multidisciplinary approach to validate the hypothesis. Molecular biological techniques (Western blot techniques, RT-PCR), biochemical analysis, and other detailed behavior testing might have added more value to the article.  I encourage the authors to think of generating more findings and figures.

R: Thanks for the suggestion. We are in alignment with Your comments regarding additional functional studies, which, should be required for the validation study. We have already written but now we emphasized the concept. Taken in consideration your indication, we consequently, we modified the type of article in “Case Report”.

Result section:

Line 108: "33o", I do not understand the symbol after 33.

R: We are sorry for the mistake. We corrected the number as 33rd.

Figures: All figures need more details. For example, how the figures are obtained, at what resolution. What are the findings the authors saw in each figure should be mentioned in detail in the figure legends.

R: Thanks for the suggestion. More details were added in the Figure Description.

Line 117-118: It would be useful to provide those data for each test, although the values are not significant.

R: This data was collected at birth in another hospital and therefore we are not able to produce the exact values, but they were reported as normal in the discharge summaries at birth so we decided to report them.

Line 156-159: Whenever possible, Presenting the data in the Tables looks much better than this text language.

R: Thanks for the suggestion. We added a table resumed the patient’s anthropometric data.

Figure 2: The scaling introduced herein in Figure 2 overlaps with the original EEG traces. I encourage you not to overlap it with the original traces. You may introduce an arrowhead for those signals that the readers need to focus on to observe the findings. Please also define those square-shaped structures within the figure itself at the bottom of the x-axis if they signify anything.

R: Thanks for the suggestion. We increased the transparency of the scale to not block the view of the EEG trace. Moreover, we have made sure that scale was in a not significant point of the EEG trace. According with your suggestion the EEG abnormalities was marked. 

Conclusion section:

Line 318-319: The opening sentence in the conclusion is not a conclusive statement. Please rewrite more appropriately.

R: Thanks for the suggestion. According to your suggestion we revised the conclusion.

Line 323: "As was reported": Please make sure that this type of grammatical issue is taken care of throughout the manuscript.

R: Thanks for the suggestion. The phrase “as was reported” was grammatically checked in the whole manuscript, as suggested.

I also see at the ending section of the introduction, that there are several one or two-sentence paragraphs, please try to merge those relevant sentences together.

R: Thanks for the suggestion. The few-sentence paragraphs were merged as suggested.

Overall, the manuscript has many grammatical errors and spacing error issues, which must be fixed. Only 4 figures are not sufficient (The first figure might not need to be the patient's face but the list of the symptoms in the box may be useful). The authors need to investigate the research problem using a more interdisciplinary approach and the sample size is very low. This article may be a better fit for the case study.

R: Thanks for the suggestion.  We revised the entire manuscript to fix the grammatical errors along the text. According with your suggestion we changed the type of the manuscript into a “Case Report”.

Round 2

Reviewer 1 Report

Comments and Suggestions for Authors

(Line 36 - 38)  The study carried out by Poke et al. [3] on DEEs epidemiology, emphasized how challenging iT is quantifying DEEs patients as many children cannot be classified in any known epilepsy syndromes. 

(Line 78) STX1A is pivotal in calcium-dependent exocytosis and endocytosis of hormones and neurotransmitters, a role highlighted [16]. SENTENCE NEEDS FIXING.

I Have read the Introduction 3 times, and I still do not understand what is DEE, the roles of the SNARE complex, STX1A, SNAP25, STXBP6 and how these components play a role in ASD, ID and other human diseases. There is also no AIM specified in the Introduction and how this research aims to address DEE. 

(Line 104-105) The 7-year-old girl was the second daughter of a healthy, non-consanguineous parents. Intrauterine growth restriction (IUGR) and reduced fetal movements were observed during pregnancy.

What was the reason for incrementing the valproic acid? Is this safe for a child this age? What confounding factors could this lead to?

Would be helpful to have included a picture of (MRI) performed at 22 months.

What was the dosage of levetiracetam and what was the motivation for this treatment? Could this not have any adverse effects on the child?

Can mutations in the STX1B, STX1A and STXBP6 gene be determined during pregnancy? 

(Line 273 - 274) herefore, the STRING analysis (data not shown) revealed a robust interaction between STXBP6 and SNAP25 genes, reinforcing the well-documented association of SNAP25 with autism. 

(Line 336 - 338) These findings provide valuable insights into the potential involvement of STXBP6 gene in the pathogenesis of neurological conditions, expanding on the body of evidence on SNAREopathies.

Comments on the Quality of English Language

English language needs to be reviewed by an English editor. There are still quite a number of typos, sentences need to be reconstructed to make sense and punctuation needs correcting.

Author Response

Dear Reviewer,

I would like to thank you for your valued comments and suggestions to the manuscript. As requested, we made all the necessary changes in our manuscript to address your concerns. The main changes are highlighted in yellow in the manuscript. According to the changes made in the revised manuscript and the responses provided below, I hope you will consider the manuscript suitable for publication. If there are any further questions, please feel free to contact me.

Sincerely,

Dr. Francesco Cali.

REVIEWER 1

(Line 36 - 38) The study carried out by Poke et al. [3] on DEEs epidemiology, emphasized how challenging iT is quantifying DEEs patients as many children cannot be classified in any known epilepsy syndromes.

A:  Thanks for the suggestion. We corrected as you indicated.

(Line 78) STX1A is pivotal in calcium-dependent exocytosis and endocytosis of hormones and neurotransmitters, a role highlighted [16]. SENTENCE NEEDS FIXING.

A:  Thanks for the suggestion. The sentence was rewritten.

I Have read the Introduction 3 times, and I still do not understand what is DEE, the roles of the SNARE complex, STX1A, SNAP25, STXBP6 and how these components play a role in ASD, ID and other human diseases. There is also no AIM specified in the Introduction and how this research aims to address DEE. 

A:  Thanks for the suggestion. We clarified the concept of DEE in the introduction as defined today by ILAE. Moreover, we underlined again the correlation between SNARE complex and neurodevelopmental impairment that can lead to autism or intellectual disability. The exact mechanism in which mutations of SNARE complex can disrupt neurodevelopment remains largely unknown and goes beyond the scope of this case report. Finally, we underlined that the aim of our case report is to report the first case of a DEE associated with a de novo STXBP6 mutation (see lines 123-125).

(Line 104-105) The 7-year-old girl was the second daughter of healthy, non-consanguineous parents. Intrauterine growth restriction (IUGR) and reduced fetal movements were observed during pregnancy.

A:  Thanks for the suggestion. We corrected as you indicated.

What was the reason for incrementing the valproic acid? Is this safe for a child this age? What confounding factors could this lead to?

A:  Thanks for the suggestion. Every antiepileptic drug treatment is started gradually. We gradually reached a dosage of 150 mg of valproic acid (VPA) twice a day due to the lack of optimal control of the seizures. We specified this in the text. Valproic acid (and levetiracetam) is a common antiepileptic drug used for the treatment of early onset epilepsy and it is generally well tolerated.

Would be helpful to have included a picture of (MRI) performed at 22 months.

 A:  Thanks for the suggestion. We added an axial MRI T2-image at 22 months showing simplified cortical spinning pattern, mild ventriculomegaly and periventricular white substance thinning.

What was the dosage of levetiracetam and what was the motivation for this treatment? Could this not have any adverse effects on the child?

A:  levetiracetam (LVT) was introduced because the patient was not seizure free.  LVT is one of the most common drugs used in childhood and it has an excellent tolerability profile and it is generally well tolerated. Thanks to the antiepileptic drugs combination (VPA+LVT), the patient has been seizure free for several years.

Can mutations in the STX1B, STX1A and STXBP6 gene be determined during pregnancy? 

A: Yes they can, but only in theory because these genes are not included in routine prenatal genetic testing.

(Line 273 - 274) herefore, the STRING analysis (data not shown) revealed a robust interaction between STXBP6 and SNAP25 genes, reinforcing the well-documented association of SNAP25 with autism. 

A:  Thanks for the suggestion. We corrected as you indicated.

(Line 336 - 338) These findings provide valuable insights into the potential involvement of STXBP6 gene in the pathogenesis of neurological conditions, expanding on the body of evidence on SNAREopathies.

A:  Thanks for the suggestion. We corrected as you indicated.

Reviewer 2 Report

Comments and Suggestions for Authors

I am satisfied with the new version of the manuscript and changing of article type into “case study”.

Comments on the Quality of English Language

English quality is good after revision.

Author Response

Dear Reviewer,

I would like to thank you for the valued comments and suggestions to the manuscript.  If there are any further questions, please feel free to contact me.

Sincerely,

Dr. Francesco Cali.

Reviewer 3 Report

Comments and Suggestions for Authors

The article has been greatly improved at this stage to the next level.

Author Response

(The authors gave the same response as above.)
